# Review of Ethnomedicinal, Phytochemical and Pharmacological Properties of *Lannea schweinfurthii* (Engl.) Engl.

**DOI:** 10.3390/molecules24040732

**Published:** 2019-02-18

**Authors:** Alfred Maroyi

**Affiliations:** Medicinal Plants and Economic Development (MPED) Research Centre, Department of Botany, University of Fort Hare, Private Bag X1314, Alice 5700, South Africa; amaroyi@ufh.ac.za; Tel.: +27-719-600-326

**Keywords:** Anacardiaceae, ethnopharmacological, *Lannea schweinfurthii*, south, central and east Africa

## Abstract

*Lannea schweinfurthii* is a valuable medicinal plant species among different ethnic groups in tropical Africa. The aim of the current study was to review existing literature on the ethnomedicinal uses, phytochemistry and pharmacological properties of *L. schweinfurthii* in an effort to evaluate the therapeutic potential of the species. The relevant information on the ethnomedicinal uses, phytochemistry and pharmacological properties of *L. schweinfurthii* was generated from several sources including books, book chapters, theses, scientific reports and journal articles obtained from the library and internet sources such as SciFinder, Wiley, Web of Science, American Chemical Society publications, PubMed, BMC, Elsevier, Science Direct, Scielo and Scopus. Traditionally, *L. schweinfurthii* is used against reproductive system disorders, respiratory disorders, injuries, headache, blood system disorders, infections or infestations and gastro-intestinal disorders. The phytochemical compounds that have been isolated from *L. schweinfurthii* include alkaloids, anthocyanins, flavonoids, glycosides, phenols, saponins, steroids, tannins and terpenoids. The in vitro and animal studies carried out so far from the crude extracts and compounds isolated from the species exhibited acetylcholinesterase inhibitory, anti-apoptotic, antibacterial, antiviral, anti-giardial, anti-inflammatory, antioxidant, antiplasmodial, antitrypanosomal, hepatoprotective, larvicidal and cytotoxicity activities. Detailed ethnopharmacological studies emphasizing clinical and toxicological evaluations are needed to assess efficacy and safety of the species as herbal medicine.

## 1. Introduction

*Lannea schweinfurthii* (Engl.) Engl. (Figure 1) is a deciduous, small to medium-sized tree which belongs to the wild currant or Anacardiaceae family. The Anacardiaceae family consists of trees and shrubs, rarely herbaceous climbers or lianas and all members of this family are characterized by resin channels and clear resinous sap that becomes dark on drying [1]. Myricetin flavonoids and cardanols are considered the chemotaxonomic markers of the Anacardiaceae family [2]. *Lannea schweinfurthii* is commonly referred to as “tree grape”, “bastard marula”, “baster-marula” or “false marula” [3,4] because the species produces fruits and seeds which have a sweet flavour resembling those of grapes, *Vitis vinifera* L. When *L. schweinfurthii* is not in flower or fruit, this species is sometimes confused with *Sclerocarya birrea* (A. Rich.) Hochst., commonly referred to as “marula”, hence the common names “bastard marula”, “baster-marula” or “false marula”. The genus *Lannea* consists of approximately 40 species which are usually trees, shrubs, or suffrutices, occupying different habitats in sub-Saharan Africa, Arabia and tropical Asia [3,4,5,6]. Several *Lannea* species including *L. acida* A. Rich., *L. discolor* (Sond.) Engl., *L. edulis* (Sond.) Engl. and *L. microcarpa* Engl. & K. Krause are used in the treatment and management of bacterial infections, mental disorders, fever, viral infections, gastro-intestinal disorders and fungal infections in tropical Africa [7,8,9,10,11,12]. *Lannea* species are characterized by tetracyclic and pentacyclic triterpenes, flavonoids, phenolic lipids and cyclohexene derivatives and these phytochemical compounds and crude extracts of these species exhibited anthelmintic, antibacterial, antihypertensive, anti-inflammatory, antioxidant, antiplasmodial, antiprotozoal and lipoxygenase inhibition activities [10,11,12,13,14,15,16,17,18,19,20,21,22,23,24,25,26]. Similarly, *L. schweinfurthii* is an important medicinal plant species among different ethnic groups in tropical Africa [27,28,29,30,31,32,33,34], not only used for human health care but also used as veterinary ethnomedicine in Tanzania [35] and Zimbabwe [36]. *Lannea schweinfurthii* is a popular herbal medicine in local communities in tropical Africa with the bark and leaves of the species traded as herbal medicines in informal herbal medicine markets in the Mpumalanga province in South Africa [37] and Tanzania [38,39]. Although *L. schweinfurthii* is an important component of the indigenous pharmacopoeia used for primary healthcare in tropical Africa, there is dearth of information on the phytochemical and pharmacological activities of the species. Despite considerable efforts over the last decades to document the medicinal uses and active ingredients of widely used medicinal plants in tropical Africa [9,13,40,41], there is still a knowledge gap on the medicinal uses, phytochemical and pharmacological activities of many plant species used as herbal medicines in the region. Therefore, *L. schweinfurthii* is not fully researched due to fragmented information in the literature and the current review undertaken was aimed at summarizing the medicinal uses, phytochemical and pharmacological activities of the taxon so as to provide baseline data on these aspects.

## 2. Research Methodology

Information on the medicinal uses, chemistry and pharmacological properties of *L. schweinfurthii* was collected from several sources including books, book chapters, theses, scientific reports and journal articles obtained from the University of Fort Hare library and internet sources such as SciFinder, Wiley, Web of Science, American Chemical Society publications, PubMed, BMC, Elsevier, Science Direct, Scielo and Scopus. The search for this information was carried out between June to December 2018. The keywords used in the search included “ethnobotany”, “ethnomedicinal uses”, “medicinal uses”, “phytochemistry”, “biological activities”, “pharmacological properties”, “*L. schweinfurthii*”, “*L. schweinfurthii* var. *acutifoliolata* (Engl.) Kokwaro”, “*L. schweinfurthii* var. *stuhlmannii* (Engl.) Kokwaro”, “*L. schweinfurthii* and var. *tomentosa* (Engl.) Kokwaro”, the synonyms of the species “*Commiphora porensis* Engl.”, “*L. ambigua* Engl.”, “*L. kirkii* Burtt Davy”, “*L. stuhlmannii* (Engl.) Engl.”, “*L. stuhlmannii* (Engl.) Engl. var. *tomentosa* Dunkley”, “*Odina schweinfurthii* Engl.”, “*O. stuhlmannii* Engl.”, “*O. stuhlmannii* Engl. var. *acutifoliolata* Engl.” and “*Scassellatia heterophylla* Chiov.” and the English common names “tree grape”, “bastard marula”, “baster-marula” and “false marula”. The internet search generated 867 articles in total. After duplicate articles, publications in other languages than English and those with limited raw data were excluded, 111 articles were included in this study. These articles included 70 journal articles, books (20), nine book chapters, five theses, four scientific reports and three website sources.

## 3. Taxonomy, Distribution and Description of *L. schweinfurthii*

Three varieties of *L. schweinfurthii* are recognized on the basis of distribution, hairiness and shape of the leaves. These varieties include, var. *acutifoliolata* (Engl.) Kokwaro, var. *stuhlmannii* (Engl.) Kokwaro and var. *tomentosa* (Dunkley) Kokwaro [4,42,43,44,45,46,47,48]. The leaves of var. *stuhlmannii* are usually more or less hairless while mature leaves of var. *acutifoliolata* and var. *tomentosa* are hairy on both surfaces and var. *acutifoliolata* is confined to Kenya and Tanzania unlike the other two varieties that are widespread in south, central and east Africa [4,43,47,48]. Synonyms of *L. schweinfurthii* include *Commiphora porensis* Engl., *L. ambigua* Engl., *L. kirkii* Burtt Davy, *L. stuhlmannii* (Engl.) Engl., *L. stuhlmannii* (Engl.) Engl. var. *tomentosa* Dunkley, *Odina schweinfurthii* Engl., *O. stuhlmannii* Engl., *O. stuhlmannii* Engl. var. *acutifoliolata* Engl. and *Scassellatia heterophylla* Chiov. [4,42,43,46,47,48]. The majority of ethnobotanists, published literature, local communities and traditional healers do not distinguish *L. schweinfurthii* into the three varieties, but the species is usually treated as *L. schweinfurthii sensu lato*, and this is the same approach that has been adopted in the current study.

*Lannea schweinfurthii* is a deciduous, small to medium-sized tree, growing up to 22 m in height [4,43]. The species forms a spreading and open crown with drooping branches. The bark is grey to green in colour, hairy on young stems and becomes flaky as the branches mature. The leaves are crowded at the ends of branches, imparipinnate with one to five pairs of leaflets and a terminal leaflet. The leaflets are broadly ovate or elliptic in shape with the terminal leaflet larger than lateral leaflets [4,49]. The leaflets are pale green in colour, shiny and aromatic with entire margins. The flowers are small, unisexual on different trees in axillary spikes, creamy white to yellow-green in colour. The fruit is oblong-ellipsoid in shape, fleshy and dark reddish-brown when ripe [4].

*Lannea schweinfurthii* has been recorded in river valley forests, coastal forests, lowland dry forests, dry forests, open woodlands, wooded grasslands, grasslands, dry bushvelds, rocky outcrops, termite mounds and on sandy to gravelly soil, especially those derived from igneous rocks at elevations of up to 1820 m above sea level [4,43,49,50,51]. *Lannea schweinfurthii* has been recorded in Somalia, Kenya, Sudan, Botswana, Zimbabwe, Tanzania, South Sudan, Malawi, Namibia, Swaziland, Rwanda, South Africa, Uganda, Mozambique, Zambia and Ethiopia [3,4,42,43,44,46,49,50,51,52,53,54] (Figure 2).

## 4. Medicinal Uses of *Lannea schweinfurthii*

Herbal concoctions prepared from *L. schweinfurthii* are used to treat 53 human and animal diseases in south, central and east Africa (Table 1), and in terms of the plant part utilization (Figure 2), the roots (43.0%), bark 29.0% and leaves (23.0%) are those widely used. When bark is utilized, it is usually not specified whether the bark was derived from the roots or stems (Figure 3). The major diseases and ailments treated by *L. schweinfurthii* extracts include reproductive system and respiratory disorders recorded in three countries based on three literature records each (Figure 4), followed by injuries (recorded in three countries, four literature records), headache (three countries, six literature records), blood system disorders (four countries, six literature records), infections or infestations (five countries, 13 literature records) and gastro-intestinal disorders (six countries, 16 literature records). *Lannea schweinfurthii* is used to manage and treat the top five ailments and diseases listed by the World Health Organization [55] as the leading causes of disease burden in tropical and subtropical Africa and these include human immunodeficiency virus/acquired immune deficiency syndrome (HIV/AIDS), respiratory infections, diarrhoeal diseases and birth-related disorders (Table 1). Infectious diseases such as respiratory infections, malaria, tuberculosis, diarrhoeal diseases and HIV/AIDS are considered as a global health threat, causing the majority of mortality and morbidity with a death toll of 13.4 million per year [56]. The usage of *L. schweinfurthii* against bacterial, fungal and viral infections or infestations in south central and east Africa (Table 1, Figure 3) is not surprising as microbial infections or infestations are the world’s leading causes of premature deaths [57]. *Lannea schweinfurthii* is also popular as herbal medicines against opportunistic diseases associated with HIV/AIDS which include malaria, diarrhoea, sexually transmitted infections, tuberculosis, skin infections, cancer, oral candidiasis and persistent cough in eastern and southern Africa [27,58,59,60,61,62,63]. Herbal medicines play a vital role in primary healthcare in sub-Saharan Africa [64,65], as children in rural areas of some regions in the continent, for example in southern African often suffer from diarrhoea, bilharzia and gastro-intestinal parasites [66,67]. Eloff and McGraw [66] argued that a lack of easy access to the Western healthcare facilities and the expense of orthodox medicines result in the use of herbal medicines against diarrhoea, gastrointestinal parasites and bilharzia. Eloff and McGraw [66] also argued that parasitic infestations and infections caused by helminth parasites are prevalent in poor peri-urban, rural and marginalized areas of southern Africa and these parasites are often treated with herbal medicines. Similarly, Palombo [68] also argued that diarrhoeal disease is a major cause of mortality and morbidity worldwide, especially in children in third world countries, usually as a result of infections or infestations by protozoal parasites, viruses, fungi and bacteria. Over the years, there have been several studies that have validated the use of herbal medicines as a strategy to treat or prevent gastro-intestinal disorders, infectious diseases and malaria [31,67,69]. *Lannea schweinfurthii* is also used in multi-therapeutic applications when the bark is taken, mixed with roots of *Plectranthus barbatus* Andrews and *Solanum incanum* L. as herbal medicine for blood pressure and diarrhoea in Kenya [34]. There is therefore a need to validate the ethnomedicinal applications of *L. schweinfurthii* through phytochemical and pharmacological evaluations of both the crude extracts and compounds associated with the species.

## 5. Phytochemistry of *Lannea schweinfurthii*

Chhabra et al. [87] isolated alkaloids and anthocyanins from the stem bark of *L. schweinfurthii* while Oyugi [88] identified alkaloids, flavonoids, glycosides, phenols, saponins, steroids, tannins and terpenoids from leaves and roots of *L. schweinfurthii*. The nutritional value of *L. schweinfurthii* which included classic nutrients such as carbohydrates, fibres, lipids, proteins, micro- and macro-nutrients are listed in Table 2. The values of alkaloids, saponins, total flavonols, flavonoids and phenolics are also listed in Table 2. Research by Chaves et al. [89] revealed that alkaloids extracted from plants have anticholinesterase, antioxidant, anxiolytic, anti-inflammatory and antidepressant properties. Similarly, both flavonoids and phenolics are known to have antioxidant, anticancer, antibacterial, cardioprotective agents, anti-inflammation, immune system promoting, and skin protection from UV radiation properties [90]. Steroids isolated from plants are linked to anti-inflammatory and immune-modulating properties [91] while tannins have exhibited antioxidant, antimicrobial, anti-cancer, anti-nutritional and cardio-protective properties [92]. Therefore, the presence of some of these phytochemicals in *L. schweinfurthii* as outlined in Table 3 could be used to corroborate the medicinal uses of the species as detailed in Table 1 as well as pharmacological properties such as acetylcholinesterase inhibitory [93], anti-apoptotic [94], antibacterial [2,95,96], antiviral [95,97], anti-giardial [98], anti-inflammatory [99,100], antioxidant [2,93,101], antiplasmodial [2,33,97,102,103], antitrypanosomal [104], hepatoprotective [101], larvicidal [2] and cytotoxicity [2,33,58,97,100,101,102,104,105]. Micronutrients such as copper, zinc and manganese are known to serve as antioxidants or as essential cofactors for antioxidant enzymes and immune systems are usually weakened by lack of micronutrients [88,106]. The analyses revealed that the mean concentration of lead in leaves and roots of *L. schweinfurthii* ranged from 1.0 ppm to 1.1 ppm [88] and this content is lower than the World Health Organization maximum allowable limits of 10 mg/kg [107,108]. The presence of sufficient amounts of micronutrients in leaves and roots of *L. schweinfurthii* could contribute to human nutritional requirements for normal growth and adequate protection and treatment of various diseases.

Muithya [110] identified two terpenes, namely taraxerol **35** and taraxerone **36** from the roots of *L. schweinfurthii* while Okoth [2] isolated cardanols, cyclohexenones, cyclohexenols, flavonoids and terpenes from roots, stems and leaves of the species (Table 3). Sobeh et al. [88] identified flavonoids, namely quinic acid **22**, malic acid **23**, caffeoylquinic acid **24**, epicatechin **26**, epicatechin gallate **27**, feruloylquinic acid **28**, ligustroside **29**, procyanidin dimer mono gallate **30** and rutin **31** from the bark of *L. schweinfurthii* (Table 3). Yaouba et al. [100] identified a cardanol, 3-((E)-nonadec-16′-enyl) phenol **6**, two cyclohexenols, namely 1-((E)-heptadec-14′-enyl)cyclohex-4-ene-1,3-diol **12** and 1-[pentadec-12′(E)-enyl] cyclohex-4-en-1,3-diol **17** and a flavonoid, catechin **23** from the roots of *L. schweinfurthii* (Table 3). Some of the documented medicinal uses of *L. schweinfurthii* could be attributed to the phytochemical compounds isolated from the species as these compounds exhibited antibacterial, antioxidant, antiplasmodial, larvicidal and cytotoxicity activities. Okoth [2] evaluated antibacterial activities of a mixture of compounds **7**, **8**, **9**, **10**, **11** and **27** isolated from the species against *Salmonella typhimurium*, *Escherichia coli*, *Enterococcus faecalis*, *Enterococcus faecium*, *Psudomonus aeruginosa* and *Staphylococcus aureus* using disc diffusion method with penicillin, erythromycin, vancomycin and cefuroxime as positive controls. The compound **27** was active against *Salmonella typhimurium*, *Enterococcus faecalis*, *Enterococcus faecium*, *Escherichia coli*, *Psudomonus aeruginosa* and *Staphylococcus aureus* with zone of inhibition ranging from 8 mm to 22 mm while a mixture of compounds **7**, **8**, **9**, **10** and **11** exhibited activities against *Enterococcus faecalis, Enterococcus faecium* and *Staphylococcus aureus* with 10 mm zone of inhibition. Okoth [2] determined the antioxidant activities of the compounds **25**, **26**, **27** and **31** isolated from *L. schweinfurthii* roots using the 2,2-diphenyl-1-picryl hydrazyl (DPPH) radical scavenging assay with ascorbic acid as the standard drug. The compounds showed activities with half maximal inhibitory concentration (IC_50_) values ranging from 7.3 µg/mL to 22.8 µg/mL which were comparable to IC_50_ value of 5.0 µg/mL exhibited by ascorbic acid, the standard drug. Okoth [2] evaluated anti-plasmodial activities of a mixture of compounds **7**, **8**, **9**, **10**, **11** and **27** isolated from *L. schweinfurthii* against chloroquine sensitive (D6) and resistant (W2) *Plasmodium falciparum* with mefloquine and chloroquine as positive controls. The compound **27** exhibited activities with IC_50_ values of 2.8 µg/mL and 2.1 µg/mL against D6 and W2, respectively while the mixture of compounds exhibited IC_50_ values of 30.0 µg/mL and 24.1 µg/mL against D6 and W2, respectively. Okoth [2] evaluated the larvicidal activities of a mixture of compounds isolated from *L. schweinfurthii*, that is, compounds **2, 3, 4** and **5** (mixture 1), compounds **1**, **2**, **3**, **4**, **5** and **6** (mixture 2), compounds **7**, **8**, **9**, **10** and **11** (mixture 3), compounds **13**, **17**, **18** and **21** (mixture 4) and compounds **14**, **15**, **16**, **17**, **18** and **19** (mixture 5) against the malaria vector, *Anopheles gambiae* which were bio-assayed following WHO susceptibility protocols [111]. The compound mixtures exhibited activities with half maximal lethal concentration (LC_50_) values ranging between 6.3 µg/mL to 240.4 µg/mL which were higher than 3.1 µg/mL exhibited by pylarvex, the standard drug used as control. Okoth [2] evaluated the cytotoxicity activities of compounds lupenone **32**, taraxerol **35** and taraxerone **36** as well as a mixture of compounds isolated from *L. schweinfurthii*, that is, compounds **2**, **3**, **4** and **5** (mixture 1), compounds **1**, **2**, **3**, **4**, **5** and **6** (mixture 2), compounds **7**, **8**, **9**, **10** and **11** (mixture 3), compounds **13**, **17**, **18** and **21** (mixture 4) and compounds **14**, **15**, **16**, **17**, **18** and **19** (mixture 5) on a Chinese Hamster Ovarian mammalian cell-line using the 3-(4,5-dimethylthiazol-2-yl)-2,5-diphenyl tetrazolium bromide (MTT) calorimetric assay. Mixtures 1, 2, 4 and 5 were non-toxic with IC_50_ values ranging from 80.5 µg/mL to >100 µg/mL relative to 0.07 µg/mL exhibited by the standard emetine. However, mixture 3 could be considered to be potentially toxic as it exhibited IC_50_ values of 5.1 µg/mL. Lupenone **31**, taraxerol **35** and taraxerone **36** exhibited activities with IC_50_ values of 1.0 µg/mL, 42.2 µg/mL and 56.2 µg/mL [2]. Yaouba et al. [100] evaluated cytotoxicity activities of compounds **6**, **12**, **17** and **25** isolated from *L. schweinfurthii* using the MTT assay on the mammalian African monkey kidney (Vero) and DU-145 prostate cancer cell lines. The compound **6** showed activities against the Vero cell line with half maximal cytotoxic concentration (CC_50_) values of 16.1 μg/mL [100]. 

## 6. Pharmacological Properties and Safety Evaluation of *L. schweinfurthii*

The different plant parts of *L. schweinfurthii* have been subjected to a series of biological activities, specifically acetylcholinesterase inhibitory [93], anti-apoptotic [94], antibacterial [2,95,96], antiviral [95,97], anti-giardial [98], anti-inflammatory [99,100], antioxidant [2,93,101], antiplasmodial [2,33,97,102,103], antitrypanosomal [104], hepatoprotective [101], larvicidal [2] and cytotoxicity [2,33,58,97,100,101,102,104,105] (Table 4). The majority of the documented biological activities are in vitro-based assays although a few, such as anti-inflammatory [100], antiplasmodial [102], hepatoprotective [101] and toxicity [102] which have utilized in vivo models (Table 4).

### 6.1. Acetylcholinesterase Inhibitory Activities

Adewusi and Steenkamp [93] determined the acetylcholinesterase inhibitory (AChEl) of the ethyl acetate and methanol extracts of *L. schweinfurthii* roots using the Ellman’s colorimetric method with galantamine as a positive control. Acetylcholinesterase inhibitory activities were observed to be dose-dependent with IC_50_ values of 0.0003 mg/mL for the ethyl acetate extract [93]. Further research is required on this aspect as these preliminary findings support the traditional use of *L. schweinfurthii* for treating and managing fits in Zambia [29], mental disorders in Mozambique [28], headache in Kenya, South Africa and Tanzania [27,51,52,75,76,78] and as sedative in South Africa and Swaziland [8,82,85].

### 6.2. Anti-Apoptotic Activities

Seoposengwe et al. [93] evaluated anti-apoptotic activities of ethyl acetate and methanol extracts of *L. schweinfurthii* root bark in SH-SY5Y neuroblastoma cells. In vitro assays employed included intracellular redox state, reactive oxygen species (ROS), intracellular glutathione content, mitochondrial membrane potential (MMP) and caspase-3 activity of the extracts. The methanol extract produced LC_50_ value of 78.9 μg/mL, while ethyl acetate extract produced LC_50_ value of 36.0 μg/mL. Both the methanol and ethyl acetate extracts demonstrated cytoprotective properties in cells exposed to concentrations of 10 nM of rotenone, however, the same extracts did not show any significant cytoprotective effects at rotenone concentrations of 50 nM and 100 nM. Both the methanol and ethyl acetate extracts countered the decrease in intracellular ROS caused by rotenone exposure. The extracts produced significant increases in glutathione content in a dose-dependent manner. Pre-treating cells with the extracts caused a further reduction in MMP, when compared to rotenone exposure alone. The extracts significantly reduced rotenone-induced caspase-3 activity with methanol extract being more effective than ethyl acetate extract [93].

### 6.3. Antibacterial Activities

Maregesi et al. [95] evaluated antibacterial activities of aqueous and methanol extracts of *L. schweinfurthii* stem bark against *Bacillus cereus*, *Staphylococcus aureus*, *Escherichia coli*, *Pseudomonas aeruginosa*, *Klebsiella pneumoniae* and *Salmonella typhimurium* using a liquid dilution method with ampicillin and rifampicin as positive controls. The methanol extract exhibited activities against *Bacillus cereus* with MIC and MBC value of 500 μg/mL while aqueous extract exhibited MIC and MBC value of 1000 μg/mL against *Bacillus cereus* [95]. Okoth [2] evaluated antibacterial activities of hexane and methanol extract of *L. schweinfurthii* roots and stems against *Salmonella typhimurium*, *Escherichia coli*, *Enterococcus faecalis*, *Enterococcus faecium*, *Psudomonus aeruginosa* and *Staphylococcus aureus* using disc diffusion method with penicillin, erythromycin, vancomycin and cefuroxime as positive controls. The hexane extract of the roots was active against *Enterococcus faecalis* and *Enterococcus faecium* with 10 mm zone of inhibition while methanol extract of both the roots and stems exhibited activities against *Salmonella typhimurium*, *Enterococcus faecalis*, *Enterococcus faecium*, *Psudomonus aeruginosa* and *Staphylococcus aureus* with zone of inhibition ranging from 8 mm to 15 mm [2]. Similarly, Lall et al. [96] evaluated antibacterial activities of ethanol extract of *L. schweinfurthii* root bark against *Mycobacterium smegmatis* and *Propionibacterium acnes* using micro-dilution technique with ciproflaxin and tetracycline as positive controls. The extract exhibited weak activities against *Mycobacterium smegmatis* and *Propionibacterium acnes* with MIC values >1000 µg/mL and 125 µg/mL, respectively against 0.31 µg/mL and 0.78 µg/mL exhibited by the positive controls ciproflaxin and tetracycline, respectively [96]. There is a need for more research as *L. schweinfurthii* leaf, root and stem bark extracts are traditionally used as herbal medicines against bacterial infections such as boils [27,29,60], carbuncles [27], cellulitis [31], cough [75], gastro-intestinal problems [9,27,28,29,32,33,51,52,60,73,74,75,76,77,78,79], gonorrhoea [28,34,61,62,63], skin infections [59,62,75], syphilis [31,61,62], tuberculosis [27,28], tussis [28] and venereal diseases [70,75,80].

### 6.4. Antiviral Activities

Maregesi et al. [95] evaluated antiviral activities of methanol extract of *L. schweinfurthii* stem bark against Herpes Simplex Virus type 1, Vesicular Stomatitis Virus T2, Coxsackie B2 and Semliki Forest A7 using of 50% end point titration technique (50% EPPT) with acyclovir as a positive control. The extract showed strong antiviral activities with reduction factor (RF) values of 101 and 103 at concentrations of 25 µg/mL and 50 µg/mL, respectively against Semliki Forest virus A7 [95]. Similarly, Maregesi et al. [97] evaluated the anti-HIV activities of aqueous and 80% methanol extract of *L. schweinfurthii* stem bark against human immunodeficiency virus type 1 (HIV-1, III_B_ strain) and type 2 (HIV-2, ROD strain) using the micro dilution assay. The 80% methanol extract exhibited the best activities with IC_50_ values ranging from 7.1 µg/mL to 9.9 µg/mL while aqueous extract exhibited weaker activities with IC_50_ values ranging from 53.2 µg/mL to 89.4 µg/mL [97]. There is need for further research as *L. schweinfurthii* bark leaves and roots are traditionally used as herbal medicines against viral infections and coughs in Kenya [75], herpes simplex and zoster in Namibia and Zambia [59,62], smallpox in Zambia [29], tussis in Mozambique [28] and venereal diseases in Kenya and Mozambique [70,75,80].

### 6.5. Anti-giardial Activities

Johns et al. [98] evaluated anti-giardial activities of methanol extract of *L. schweinfurthii* root bark using a bioassay of the growth inhibition of *Giardia lamblia* trophozoites with metronidazole as a positive control. The methanolic extract were lethal or inhibited growth of *Giardia lamblia* trophozoites at 1000 ppm [98].

### 6.6. Anti-inflammatory Activities

Lawal et al. [99] evaluated anti-inflammatory activities of acetone extracts of *L. schweinfurthii* bark against lipoxygenase (15-LOX) enzyme. The extract exhibited activities with IC_50_ value of 43 μg/mL [99]. Yaouba et al. [100] evaluated in vivo anti-inflammatory activities of methanol extract of *L. schweinfurthii* roots using the carrageenan-induced rat paw edema assay in adult Wistar rats, where 200 mg/kg body weight of the extract was administrated orally to different groups of rats with indomethacin (10 mg/kg) as the positive control. The extract showed moderate activities at 60 min and 120 min post-carrageenan administration and showed the smallest increase in paw volume observed at any time point [100]. The observed in vivo anti-inflammatory activity of *L. schweinfurthii* extract of the roots partially support the reported traditional use of *L. schweinfurthii* the relief of pain and inflammation such as abdominal pain in Tanzania [27], body pains in South Africa [83], sores in Swaziland and Tanzania [27,58,82] swellings in Kenya [75] and wounds in Tanzania [81]. 

### 6.7. Antioxidant Activities

Adewusi and Steenkamp [93] determined the antioxidant activities of the ethyl acetate and methanol extracts of *L. schweinfurthii* roots using the DPPH and 2,2´-azinobis-3-ethylbenzothiazoline-6-sulfonic acid (ABTS) radical scavenging assays. The methanol extract of both the DPPH and ABTS methods showed activities with IC_50_ values of 0.01 mg/mL and 0.004 mg/mL, respectively [93]. Okoth [2] determined the antioxidant activities of the methanol extract of *L. schweinfurthii* roots using the DPPH radical scavenging assay with ascorbic acid as the standard. The extract showed activities with IC_50_ value of 22.8 µg/mL which was comparable to IC_50_ value of 5.0 µg/mL exhibited by ascorbic acid, the standard drug [2]. Sobeh et al. [101] evaluated antioxidant activities of methanol extract of *L. schweinfurthii* bark using the DPPH radical scavenging and ferric reducing antioxidant power (FRAP) assays. The extract of the DPPH method showed activities with EC_50_ value of 5.6 μg/mL which was comparable to EC_50_ value of 2.9 µg/mL exhibited by ascorbic acid, the standard drug. The extract of the FRAP method showed activities with 18.3 mM FeSO_4_ equivalent/mg of sample which was comparable to 24.0 mM FeSO_4_ equivalent/mg of sample exhibited by quercetin, the standard drug. In in vivo experiments where antioxidant capacity (TAC) of the extract was examined in male Wistar rats treated with toxic d-galactosamine where TAC was determined as well as lipid peroxidation product, malondialdehyde (MDA) in liver tissues as markers for oxidative stress. The extract was able to counteract the d-galactosamine-intoxication and caused an increase in TAC of liver tissues, the low dose levels of 100 mg/kg body weight were able to attenuate the MDA increase [101].

### 6.8. Antiplasmodial Activities

Gathirwa et al. [95] evaluated the antiplasmodial activities of aqueous and methanol extracts of *L. schweinfurthii* stem bark using an in vitro semi-automated micro-dilution assay technique that measures the ability of the extracts to inhibit the incorporation of [G-^3^H]hypoxanthine into the malaria parasite *Plasmodium falciparum* chloroquine sensitive (D6) and chloroquine resistant (W2). The in vivo anti-plasmodial activities of aqueous and methanol extracts of *L. schweinfurthii* roots were evaluated on male Balb C mice. The *Plasmodium berghei* strain maintained by serial passage of infected blood through interperitonial injection was used based on the four-day suppressive test. Gathirwa et al. [95] also evaluated the in vitro and in vivo drug interactions of aqueous and methanol stem bark extracts of *L. schweinfurthii* in combination with *Boscia salicifolia* Oliv., *Searsia natalensis* (Bernh. ex C. Krauss) F. A. Barkley, *Turraea robusta* Gürke and *Sclerocarya birrea* (A. Rich.) Hochst. against W2. The methanol and aqueous extracts exhibited low activities with IC_50_ values of 36.3 µg/mL and 75.9 µg/mL, respectively against W2. The aqueous and methanol extracts exhibited moderate activities with IC_50_ values of 10.6 µg/mL and 11.4 µg/mL, respectively against D6. The methanol and water extracts were active in interperitonial injection treatment with chemo-suppression of 91.4% and 83.1%, respectively and the chemo-suppression of malaria parasites by extracts was not significantly different from that of chloroquine. Moderate to weak synergism was observed in most combinations of *L. schweinfurthii* and *Turraea robusta* except towards equal proportion of each extract (50:50) where moderate additive interaction was noted. The interaction of *L. schweinfurthii* and *Searsia natalensis* was additive at all tested ratios. All combinations of *L. schweinfurthii* and *Boscia salicifolia* extracts gave additive interaction except at 70:30 where synergistic behaviour was observed and at a high amount of *L. schweinfurthii* while blended with *Sclerocarya birrea*, additive behaviour was observed which changed to antagonism at the high amount of the latter. The combination of aqueous extracts of *Turraea robusta* and *L. schweinfurthii* (10:90) also exhibited antagonistic interaction with chemo-suppression of the combination (57.5%) being lower than that of the single extracts (63.8 and 83.1%, respectively). The combination of *Turraea robusta* and *L. schweinfurthii* exhibited high chemosupression of 94.4%. Testing these combinations in vivo demonstrated enhanced anti-malarial activities compared to the single *L. schweinfurthii* extracts with some giving chemosuppression close to that of chloroquine. The mean survival times of mice treated with blends of *L. schweinfurthii* and *Boscia salicifolia* were not significantly different from the control group treated with chloroquine [95]. Maregesi et al. [97] evaluated the antiplasmodial activities of 80% methanol extracts of *L. schweinfurthii* stem bark against *Plasmodium falciparum* using twofold serial dilutions method with chloroquine diphosphate as a positive control. The extracts exhibited activities with IC_50_ values ranging from 62.3 µg/mL to 125 µg/mL and MIC value of 125 µg/mL [97]. Gathirwa et al. [33] evaluated in vitro antiplasmodial activities of methanol leaf extracts of *L. schweinfurthii* by measuring the ability of the extracts to inhibit the incorporation of radio-labelled hypoxanthine into the malaria parasite *Plasmodium falciparum* chloroquine sensitive (D6) and chloroquine resistant (W2) with chloroquine as a positive control. The extracts were tested singly and then in combination with leaves of *Searsia natalensis* using the standard fixed ratio analysis to evaluate synergism. In vivo bioassay was done in mice using Peter’s 4-days suppressive test. The extracts showed activities with IC_50_ values of 38.9 µg/mL and 54.2 µg/mL against D6 and W2, respectively. When tested in vivo in a mouse model, *L. schweinfurthii* extract alone and in combination with *Searsia natalensis* showed percent parasite clearance and chemo-suppression of 83.5% and 87.7%, respectively which was comparable to 95.2% exhibited by the positive control, chloroquine (5 mg/kg). Evaluating the effect of combining *L. schweinfurthii* extracts with *Searsia natalensis* leaves against a multidrug resistant W2 revealed synergism with the sum of fractional inhibition concentration (SFIC) values ranging from 0.4 to < 1.0 [33]. Muthaura et al. [103] evaluated antiplasmodial activities of aqueous and methanol extracts of *L. schweinfurthii* stem bark against chloroquine sensitive (D6) and resistant (W2) Plasmodium falciparum using the semi-automated micro-dilution technique that measures the ability of the extracts to inhibit the incorporation of (G-^3^H) hypoxanthine into the malaria parasite. The aqueous and methanol extracts exhibited low activities with IC_50_ values of 36.3 µg/mL and 75.9 µg/mL, respectively against W2. The aqueous and methanol extracts exhibited moderate activities with IC_50_ values of 10.6 µg/mL and 11.4 µg/mL, respectively against D6 [103]. These documented antiplasmodial activities support the use of *L. schweinfurthii* bark as herbal medicine against malaria in Tanzania [58].

### 6.9. Antitrypanosomal Activities

Nibret et al. [104] evaluated in vitro antitrypanosomal activities of dichloromethane and methanol extracts of *L. schweinfurthii* roots using quantitative colorimetric assay via the oxidation–reduction indicator resazurin against *Trypanosoma brucei brucei* with diminazene aceturate as the standard drug. The extracts exhibited activities with IC_50_ values of 22.2 µg/mL and 44.2 µg/mL against dichloromethane and methanol extracts, respectively which were higher than 0.09 µg/mL exhibited by diminazene aceturate, the standard drug [104]. 

### 6.10. Hepatoprotective Activities

Sobeh et al. [101] evaluated hepatoprotective activities of *L. schweinfurthii* bark extracts in male Wistar rats treated with the toxic d-galactosamine that causes liver damage through evaluation of both the biochemical and histopathological changes in rats. The extracts reduced the elevated levels of serum aspartate aminotransferase (AST) and total bilirubin and significantly attenuated the deleterious histopathologic changes in liver after d-galactosamine-intoxication [101].

### 6.11. Larvicidal Activities

Okoth [2] evaluated the larvicidal activities of ethyl acetate, hexane and methanol extracts of *L. schweinfurthii* leaf, root and stem against the malaria vector, *Anopheles gambiae* which was bio-assayed following WHO susceptibility protocols [111]. The extracts exhibited activities with LC_50_ values ranging between 6.3 µg/mL to 240.4 µg/mL which were higher than 3.1 µg/mL exhibited by pylarvex, the standard drug used as control [2]. The results suggest that the investigated leaf, root and stem extracts of *L. schweinfurthii* are promising as larvicides against *Anopheles gambiae* mosquitoes and could be useful leads in the search for new and biodegradable plant derived larvicide products.

### 6.12. Cytotoxicity and Toxicity Activities

Moshi et al. [58] evaluated toxicity activities of methanol extract of *L. schweinfurthii* stem using the brine shrimp lethality test. The extract exhibited activities with LC_50_ value of 67.9 μg/mL [58]. Gathirwa et al. [102] evaluated cytotoxicity activities of methanol extract of *L. schweinfurthii* roots on Vero cells using the plaque reduction assay. The methanol extract exhibited weak activities with CC_50_ value of 225.3 µg/mL and selectivity index of 6.2 against W2 [101]. Gathirwa et al. [101] evaluated in vivo acute toxicity activities of methanol root extract of *L. schweinfurthii* using Swiss mice. The groups of five starved mice were given increasing dosages of test sample, 250 mg/kg body weight, 500 mg/kg body weight, 750 mg/kg body weight, 1000 mg/kg body weight, 1500 mg/kg body weight and 5000 mg/kg body weight orally and the number of deaths occurring within 24 h to 48 h were noted. In the acute toxicity assay, no deaths were observed at the highest concentration tested and the LD_50_ values for both extracts were above 5000 mg/kg body weight when tested by oral administration [102]. Nibret et al. [104] evaluated cytotoxicity of dichloromethane and methanol extracts of *L. schweinfurthii* roots against the human cancer cell line HL-60 cells with diminazene aceturate as the standard drug. The dichloromethane extract was the most cytotoxic extract with IC50 value of 27.2 µg/mL and selectivity index of 1.2 which was much higher than IC_50_ value of >128.9 µg/mL and selectivity index value of >1464.0 exhibited by diminazene aceturate, the standard drug [104]. Maregesi et al. [97] evaluated cytotoxicity activities of aqueous and 80% methanol extracts of *L. schweinfurthii* stem bark on MT-4 cells. Both extracts exhibited activities with CC_50_ values of 72.3 µg/mL and >125 µg/mL for 80% methanol and aqueous extracts, respectively [97]. Gathirwa et al. [33] evaluated cytotoxicity activities of aqueous extracts of *L. schweinfurthii* leaves using growth inhibition of Vero E6 cells. The extract exhibited mild cytotoxicity with CC_50_ values of 76 µg/mL and a selectivity index of 1.4 [33]. Adewusi et al. [105] evaluated cytotoxicity activities of methanol extracts of *L. schweinfurthii* roots against SH-SY5Y (human neuroblastoma) cells which were untreated, as well as toxically induced with amyloid-beta peptide (Aβ) using the MTT and neutral red uptake assays. Exposure of the SH-SY5Y cells to Aβ25-35 at 2.5 µM, 5 µM, 10 µM and 20 µM for 72 h, resulted in a concentration-dependent decrease in cell survival. The extracts exhibited low cytotoxicity with IC_50_ values exceeding 100 µg/mL from both the neutral red and MTT assays. The pre-treatment with extract resulted in an optimum dose for inhibition of Aβ25-35 induced cell death at 25 µg/mL, while still maintaining 80% viability [105]. Yaouba et al. [100] evaluated cytotoxicity activities of root extracts of *L. schweinfurthii* using the assay on the mammalian African monkey kidney (Vero) and DU-145 prostate cancer cell lines. The extracts showed activities with CC_50_ values of 7.4 μg/mL and 74.0 μg/mL against the Vero cell line and DU-145 prostate cancer cell lines, respectively [100]. 

## 7. Conclusions

In this article, the ethnomedicinal uses, phytochemistry and pharmacological activities of *L. schweinfurthii* were reported emphasizing its acetylcholinesterase inhibitory, anti-apoptotic, antibacterial, antiviral, anti-giardial, anti-inflammatory, antioxidant, antiplasmodial, antitrypanosomal, hepatoprotective, larvicidal and cytotoxicity activities. The diverse uses of *L. schweinfurthii* among different ethnic groups in tropical Africa and the scientific evidence of its phytochemistry and pharmacological properties indicate therapeutic potential of the species. However, there is need for clinical evaluation of crude extracts and compounds isolated from the species using in vivo models. Therefore, further studies of *L. schweinfurthii* should focus on comprehensive phytochemical analyses of the species and to more accurately define its biological activities so that biological activities can be correlated to phytochemical components. Additionally, new biological evaluations are still needed to scientifically validate some of its ethnopharmacological applications so that the species can be used as a future resource for disease treatment and management. *Lannea schweinfurthii* is also not fully evaluated regarding its safety as herbal medicine. Further studies should therefore, evaluate the toxicity of the species on humans.

## Figures and Tables

**Figure 1 molecules-24-00732-f001:**
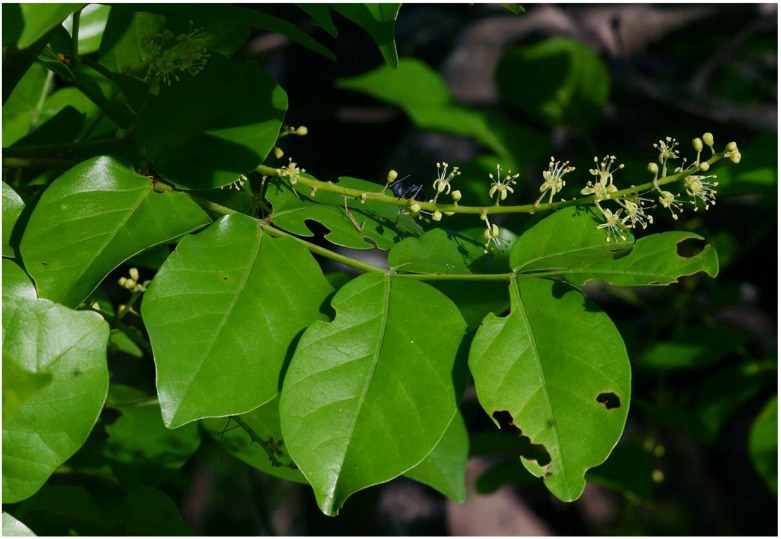
*Lannea schweinfurthii*: a branch showing leaves and flowers (photo: BT Wursten).

**Figure 2 molecules-24-00732-f002:**
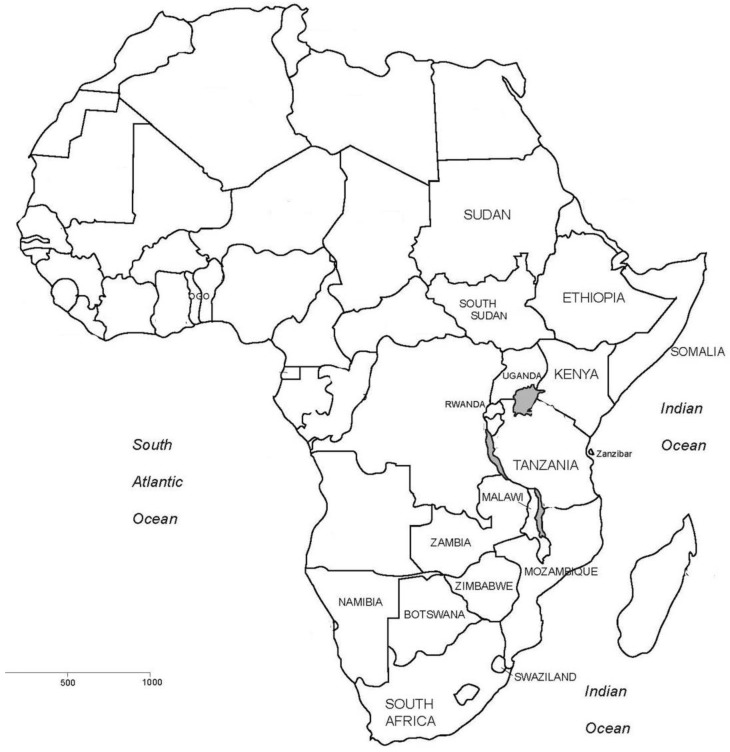
Countries in which *Lannea schweinfurthii* has been recorded in south, central and east Africa.

**Figure 3 molecules-24-00732-f003:**
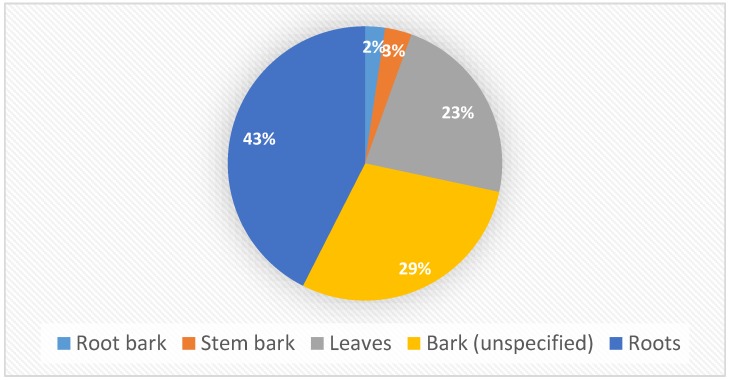
Different plant parts of *Lannea schweinfurthii* used as herbal medicines in south, central and east Africa.

**Figure 4 molecules-24-00732-f004:**
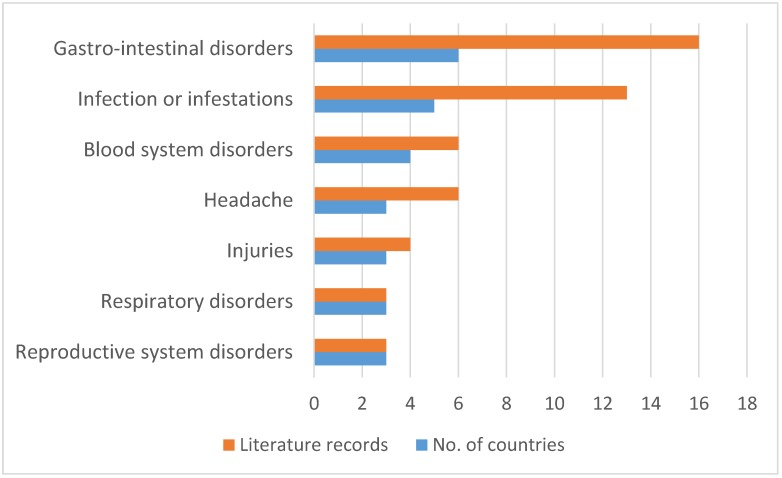
Major diseases treated by *Lannea schweinfurthii* extracts in south, central and east Africa.

**Table 1 molecules-24-00732-t001:** Medicinal uses of *Lannea schweinfurthii* in south, central and east Africa.

Medicinal Use	Parts of the Plant Used	Country	References
Animal diseases (coccidiosis, corridor disease and prophylactic measure against poultry diseases)	Roots	Tanzania and Zimbabwe	[35,36]
Birth-related disorders (induce labour, pre, intra and post-partum, post-partum haemorrhage, pregnancy anaemia and retained after birth)	Bark, leaves and roots	Kenya and Tanzania	[27,70,71]
Blood pressure and diarrhoea	Bark mixed with roots of *Plectranthus barbatus* Andrews and *Solanum incanum* L.	Kenya	[34]
Blood system disorders (anaemia and blood pressure)	Bark and roots	Botswana, Kenya, Mozambique and Tanzania	[27,32,34,38,50,72]
Fever and malaria	Bark and roots	Tanzania	[27,58]
Fits	Bark and leaves	Zambia	[29]
Gastro-intestinal disorders (amoebic dysentery, diarrhoea, dysentery, stomach problems and stomachache)	Stems, leaves, bark and roots	Kenya, Mozambique, South Africa, Tanzania, Zambia and Zimbabwe	[9,27,28,29,32,33,50,51,60,73,74,75,76,77,78,79]
Headache	Leaves, bark and roots	Kenya, South Africa and Tanzania	[27,50,51,75,76,78]
Infections or infestations (abscesses, boils, carbuncles, cellulitis, gonorrhoea, herpes simplex, herpes zoster, HIV/AIDS, oral candidiasis, skin infections, skin rash, smallpox, syphilis and venereal diseases)	Stems, roots, leaves and bark	Kenya, Mozambique, Namibia, Tanzania and Zambia	[27,28,29,31,34,59,60,61,62,63,70,75,80]
Injuries (sores and wounds)	Bark and roots	South Africa, Swaziland and Tanzania	[9,27,58,81,82]
Mental disorder	Leaves	Mozambique	[28]
Pain (abdominal and body pains)	Leaves and roots	South Africa and Tanzania	[27,83]
Protective charm	Roots	Kenya and South Africa	[8,70,76,84]
Reproductive system disorders (abortifacient and sterility)	Leaves	Kenya, Tanzania and Uganda	[27,30,79]
Respiratory disorders (asthma, cough, tuberculosis and tussis)	Bark, leaves and roots	Kenya, Mozambique and Tanzania	[27,28,75]
Sedative	Roots	South Africa and Swaziland	[8,82,85]
Snake bite	Leaves and root bark	Kenya and South Africa	[8,34,86]
Stomach ulcers	Leaves and roots	Kenya and Tanzania	[79,81]
Swellings	Bark, leaves and roots	Kenya	[75]

**Table 2 molecules-24-00732-t002:** Nutritional and phytochemical composition of roots of *Lannea schweinfurthii*.

Nutritional Composition	Values	Plant Parts	Reference
Acid detergent fibre (%)	14.4	Leaves	[109]
Alkaloids (%)	1.8–4.7	Leaves and roots	[88]
Ash (%)	5.0–11.9	Leaves and roots	[88,109]
Calcium (%)	1.1	Leaves	[109]
Carbohydrates (%)	43.4–52.2	Leaves and roots	[88]
Copper (ppm)	0.9	Leaves and roots	[88]
Dry matter (%)	34.5	Leaves	[109]
Fibre (%)	0.5–1.0	Leaves and roots	[88]
In vitro dry matter digestibility (%)	61.3	Leaves	[109]
Iron (ppm)	1.4	Leaves and roots	[88]
Lead (ppm)	1.0–1.1	Leaves and roots	[88]
Lipid (%)	5.6–9.0	Leaves and roots	[88]
Manganese (ppm)	1.7–2.0	Leaves and roots	[88]
Moisture (%)	13.7–14.5	Leaves and roots	[88]
Neutral detergent fibre (%)	47.5	Leaves	[109]
Phosphorus (%)	0.3	Leaves	[109]
Protein (%)	15.4–22.0	Leaves and roots	[88,109]
Saponins (%)	10.2–16.5	Leaves and roots	[88]
Total flavonol (mg quercetin/g of extract)	17.3	Roots	[93]
Total flavonoid (mg quercetin/g of extract)	13.6	Leaves and roots	[88,93]
Total phenol (mg tannic acid/g of extract)	101.3	Roots	[93]
Total phenolic content (mg gallic acid equivalent/g of extract)	336	Bark, leaves and roots	[88,101]
Zinc (ppm)	1.2–2.6	Leaves and roots	[88]

**Table 3 molecules-24-00732-t003:** Phytochemical compounds identified from *Lannea schweinfurthii*.

No.	Compound	Molecular Formula	Plant Part	References
	**Cardanol**			
**1**	3-[heneicos-16′(Z),18′(E)-dienyl] phenol	C_27_H_44_O	Leaves, roots and stems	[2]
**2**	3-[nonadec-14′(Z),16′(E)-dienyl] phenol	C_25_H_40_O	Leaves, roots and stems	[2]
**3**	3-[heptadecyl] phenol	C_24_H_42_O	Leaves, roots and stems	[2]
**4**	3-[heptadec-12′(Z),14′(E)-dienyl] phenol	C_23_H_36_O	Leaves, roots and stems	[2]
**5**	3-[tridecyl] phenol	C_23_H_42_O	Leaves, roots and stems	[2]
**6**	3-((E)-nonadec-16′-enyl) phenol	C_25_H_42_O	Roots	[100]
	**Cyclohexenones**			
**7**	5-hydroxy-5-[tridecyl] cyclohex-2-enone	C_19_H_34_O	Leaves, roots and stems	[2]
**8**	5-hydroxy-5-[pentadecyl] cyclohex-2-enone	C_21_H_38_O	Leaves, roots and stems	[2]
**9**	5-hydroxy-5-[heptadecyl] cyclohex-2-enone	C_23_H_42_O	Leaves, roots and stems	[2]
**10**	5-hydroxy-5-[pentadec-12′(E)-enyl] cyclohex-2-enone	C_25_H_46_O	Leaves, roots and stems	[2]
**11**	5-hydroxy-5-[heptadec-14′(E)-enyl] cyclohex-2-enone	C_23_H_40_O	Leaves, roots and stems	[2]
	**Cyclohexenols**			
**12**	1-((E)-heptadec-14′-enyl)cyclohex-4-ene-1,3-diol	C_23_H_42_O_2_	Roots	[100]
**13**	1-[tridecyl] cyclohex-4-en-1,3-diol	C_19_H_36_O_2_	Leaves, roots and stems	[2]
**14**	1-[nonadecyl] cyclohex-4-en-1,3-diol	C_25_H_48_O_2_	Leaves, roots and stems	[2]
**15**	1-[heneicosyl] cyclohex-4-en-1,3-diol	C_27_H_52_O_2_	Leaves, roots and stems	[2]
**16**	1-[tricosyl] cyclohex-4-en-1,3-diol	C_29_H_56_O_2_	Leaves, roots and stems	[2]
**17**	1-[pentadec-12′(E)-enyl] cyclohex-4-en-1,3-diol	C_21_H_38_O_2_	Leaves, roots and stems	[2,100]
**18**	1-[nonadec-14′(Z),16′(E)-dienyl] cyclohex-4-en-1,3-diol	C_25_H_44_O_2_	Leaves, roots and stems	[2]
**19**	1-[heneicosen-16′(Z),18′(E)-dienyl] cyclohex-4-en-1,3 diol	C_27_H_48_O_2_	Leaves, roots and stems	[2]
**20**	1-[tridecyl] cyclohex-3-en-1,2,5-triol	C_19_H_36_O_3_	Leaves, roots and stems	[2]
**21**	1-[heptadecyl] cyclohex-3-en-1,2,5-triol	C_23_H_44_O_3_	Leaves, roots and stems	[2]
	**Cyclitol**			
**22**	Quinic acid	C_7_H_12_O_6_	Bark	[101]
	**Dicarboxylic acid**			
**23**	Malic acid	C_4_H_6_O_5_	Bark	[101]
	**Flavonoids**			
**24**	Caffeoylquinic acid	C_16_H_18_O_9_	Bark	[101]
**25**	Catechin	C_15_H_14_O_6_	Leaves, roots and stems	[2,100]
**26**	Epicatechin	C_15_H_14_O_6_	Leaves, roots and stems	[2,101]
**27**	Epicatechin gallate	C_22_H_18_O_10_	Bark, leaves, roots and stems	[2,101]
**28**	Feruloylquinic acid	C_17_H_20_O_9_	Bark	[101]
**29**	Ligustroside	C_4_H_6_O_5_	Bark	[101]
**30**	Procyanidin dimer mono gallate	C_37_H_30_O_16_	Bark	[101]
**31**	Rutin	C_27_H_30_O_16_	Leaves, roots and stems	[2]
	**Terpene**			
**32**	Lupenone	C_30_H_48_O	Leaves, roots and stems	[2]
**33**	Sitosterol	C_29_H_50_O	Leaves, roots and stems	[2]
**34**	Sitosterol glucoside	C_35_H_60_O_6_	Leaves, roots and stems	[2]
**35**	Taraxerol	C_30_H_50_O	Roots	[110]
**36**	Taraxerone	C_30_H_48_O	Roots	[110]

**Table 4 molecules-24-00732-t004:** Summary of pharmacological activities of *Lannea schweinfurthii* extracts.

Activity Tested	Extract	Plant Part	Model	Effect	Reference
Acetylcholinesterase inhibitory	Ethyl acetate	Roots	Ellman’s colorimetric	Dose dependent activity recorded with IC_50_ = 0.0003 mg/mL	[93]
Anti-apoptotic	Ethyl acetate	Root bark	Cell death by apoptosis	Exhibited activities with LC_50_ = 36.0 μg/mL	[94]
Anti-apoptotic	Methanol	Root bark	Cell death by apoptosis	Exhibited activities with LC_50_ = 78.9 μg/mL	[94]
Antibacterial	Aqueous	Stem bark	Micro-dilution assay	Exhibited activities against *Bacillus cereus* with minimum inhibitory concentration (MIC) and minimum bactericidal concentration (MBC) value of 1000 μg/mL	[95]
Antibacterial	Methanol	Stem bark	Micro-dilution assay	Exhibited activities against *Bacillus cereus* with MIC and MBC value of 1000 μg/mL	[95]
Antibacterial	Hexane	Root	Disc diffusion method	Exhibited activities against both *Enterococcus faecalis* and *Enterococcus faecium* with 10 mm zone of inhibition	[2]
Antibacterial	Methanol	Root	Disc diffusion method	Exhibited activities against both *Psudomonus aeruginosa* and *Salmonella typhimurium* with 8 mm zone of inhibition, *Enterococcus faecalis* (13 mm), *Enterococcus faecium* (15 mm) and *Staphylococcus aureus* (15 mm)	[2]
Antibacterial	Methanol	Stem	Disc diffusion method	Exhibited activities against both *Psudomonus aeruginosa* and *Salmonella typhimurium* with 8 mm zone of inhibition, *Enterococcus faecium* (12 mm), *Staphylococcus aureus* (12 mm) and *Enterococcus faecalis* (14 mm)	[2]
Antibacterial	Ethanol	Root bark	Micro-dilution technique	Exhibited activities with MIC values of >1000 µg/mL and 125 µg/mL against *Mycobacterium smegmatis* and *Propionibacterium acnes*, respectively	[96]
Antiviral	Methanol	Stem bark	50% end point titration technique	Exhibited activities with reduction factor (RF) values of 101 and 103 at concentrations of 25 µg/mL and 50 µg/mL, respectively against Semliki Forest virus A7	[95]
Antiviral	Methanol	Stem bark	50% end point titration technique	Exhibited activities with reduction factor (RF) values of 101 and 103 at concentrations of 25 µg/mL and 50 µg/mL, respectively against Semliki Forest virus A7	[97]
Antiviral	Aqueous	Stem bark	Micro-dilution technique	Exhibited activities with IC_50_ value of 53.2 µg/mL and 89.4 µg/mL against human immunodeficiency virus type 1 (HIV-1, III_B_ strain) and type 2 (HIV-2, ROD strain), respectively	[97]
Antiviral	80% methanol	Stem bark	Micro-dilution technique	Exhibited activities with IC_50_ value of 7.1 µg/mL and 9.9 µg/mL against (HIV-1, III_B_ strain and HIV-2, ROD strain, respectively	[97]
Anti-giardial	Methanol	Root bark	Growth inhibition of *Giardia lamblia* trophozoites	Extract was lethal or inhibited growth of *Giardia lamblia* trophozoites at 1000 ppm	[98]
Anti-inflammatory	Acetone	Roots	Lipoxygenase (15-LOX) inhibitory assay	Extract exhibited activities with IC_50_ value of 43 μg/mL	[99]
Anti-inflammatory	Methanol	Root	Carrageenan-induced rat paw edema method	Extracts showed moderate activities at 60 min and 120 min post-carrageenan administration	[100]
Antioxidant	Methanol	Root	2,2´-azinobis-3-ethylbenzothiazoline-6-sulfonic acid (ABTS) radical scavenging assays	Exhibited activities with IC_50_ values of 0.004 mg/mL	[93]
Antioxidant	Methanol	Roots	2,2-diphenyl-1-picryl hydrazyl (DPPH) radical scavenging assays	Exhibited activities with IC_50_ values of 0.01 mg/mL	[93]
Antioxidant	Methanol	Roots	DPPH radical scavenging assay	Exhibited activities with IC_50_ value of 22.8 µg/mL	[2]
Antioxidant	Methanol	Bark	DPPH radical scavenging assay	Exhibited activities with half maximal effective concentration (EC_50_) value of 5.6 µg/mL	[101]
Antioxidant	Methanol	Bark	Ferric reducing antioxidant power (FRAP) assays	Exhibited activities with 18.3 mM FeSO_4_ equivalent/mg sample	[101]
Antioxidant	Methanol	Bark	Antioxidant capacity (TAC) in Wistar rats	The extract increased TAC of liver tissues	[101]
Antiplasmodial	Aqueous	Stem bark	[G-^3^H] hypoxanthine incorporation assay	Extract exhibited activities with IC_50_ values of 10.6 µg/mL and 75.9 µg/mL, against Plasmodium falciparum chloroquine sensitive (D6) and chloroquine resistant (W2), respectively	[102]
Antiplasmodial	Methanol	Stem bark	[G-^3^H] hypoxanthine incorporation assay	Extract exhibited activities with IC_50_ values of 11.4 µg/mL and 36.3 µg/mL, against D6 and W2, respectively	[102]
Antiplasmodial	Aqueous	Stem bark	In vivo four-day suppressive test	Extract caused chemo-suppression of 83.1%	[102]
Antiplasmodial	Methanol	Stem bark	In vivo four-day suppressive test	Extract caused chemo-suppression of 91.4%	[102]
Antiplasmodial	Aqueous	Stem bark combined with *Turraea robusta*	In vitro drug interactions	Synergism and additive behaviour observed with sum of fractional inhibition concentration (SFIC) values of 0.6–1.7	[102]
Antiplasmodial	Aqueous	Stem bark combined with *Searsia natalensis*	In vitro drug interactions	Additive behaviour observed with SFIC values of 1.0–1.4	[102]
Antiplasmodial	Aqueous	Stem bark combined with *Boscia salicifolia*	In vitro drug interactions	Additive and synergistic behaviour observed with SFIC values of 0.9–1.3	[102]
Antiplasmodial	Aqueous	Stem bark combined with *Sclerocarya birrea*	In vitro drug interactions	Additive and antagonistic behaviour observed with SFIC values of 1.2–2.2	[102]
Antiplasmodial	Aqueous	Stem bark combined with *Turraea robusta*	In vivo drug interactions	Antagonistic behaviour observed with SFIC value 0.6, chemo-suppression of 26.8% and 57.5% after oral and interperitonial injection (i.p.) administration, respectively	[102]
Antiplasmodial	Aqueous	Stem bark combined with *Searsia natalensis*	In vivo drug interactions	Additive behaviour observed with SFIC value of 1.0, chemo-suppression of 50.7% and 71.9% after oral and i.p. administration, respectively	[102]
Antiplasmodial	Aqueous	Stem bark combined with *Boscia salicifolia*	In vivo drug interactions	Synergistic behaviour observed with SFIC value of 0.9, chemo-suppression of 61.0% and 90.6% after oral and i.p. administration, respectively	[102]
Antiplasmodial	Aqueous	Stem bark combined with *Sclerocarya birrea*	In vivo drug interactions	Synergistic behaviour observed with SFIC value of 1.2, chemo-suppression of 57.2% and 39.0% after oral and i.p. administration, respectively	[102]
Antiplasmodial	80% methanol	Stem bark	Twofold serial dilutions	Extract exhibited with IC_50_ values ranging from 62.3 µg/mL to 125 µg/mL and MIC value of 125 µg/mL	[97]
Antiplasmodial	Methanol	Leaves	[G-^3^H] hypoxanthine incorporation assay	Extract exhibited activities with IC_50_ values of 38.9 µg/mL and 54.2 µg/mL, against D6 and W2, respectively	[33]
Antiplasmodial	Methanol	Leaves combined with *Searsia natalensis*	In vitro drug interactions	Synergistic behaviour observed with SFIC values of 0.4 - <1.0	[33]
Antiplasmodial	Methanol	Leaves	In vivo four-day suppressive test	Extract caused chemo-suppression of 83.5%	[33]
Antiplasmodial	Methanol	Leaves combined with *Searsia natalensis*	In vivo four-day suppressive test	Extract caused chemo-suppression of 87.7%	[33]
Antiplasmodial	Aqueous	Stem bark	[G-^3^H] hypoxanthine incorporation assay	Extract exhibited activities with IC_50_ values of 10.6 µg/mL and 75.8 µg/mL, against D6 and W2, respectively	[103]
Antiplasmodial	Methanol	Stem bark	[G-^3^H] hypoxanthine incorporation assay	Extract exhibited activities with IC_50_ values of 11.4 µg/mL and 36.3 µg/mL, against D6 and W2, respectively	[103]
Antitrypanosomal	Dichloromethane	Roots	Colorimetric assay via the oxidation-reduction indicator resazurin	Extract exhibited activities with IC_50_ value of 22.2 µg/mL	[104]
Antitrypanosomal	Methanol	Roots	Colorimetric assay via the oxidation-reduction indicator resazurin	Extract exhibited activities with IC_50_ value of 44.2 µg/mL	[104]
Hepatoprotective	Aqueous	Stem bark	Biochemical and histopathological changes in Wistar rats	Extract reduced levels of serum aspartate aminotransferase (AST) and total bilirubin and significantly attenuated deleterious histopathologic changes in liver	[101]
Larvicidal	Ethyl acetate	Leaves, roots and stems	Larval mortality of *Anopheles gambiae*	Extracts exhibited activities with leaves exhibiting the median lethal dose (LC_50_) value of 47.9 µg/mL, stems (59.7 µg/mL) and roots (73.7 µg/mL)	[2]
Larvicidal	Hexane	Leaves, roots and stems	Larval mortality of *Anopheles gambiae*	Extracts exhibited activities with stem exhibiting LC_50_ value of 46.0 µg/mL, leaves (51.6 µg/mL) and roots (73.4 µg/mL)	[2]
Larvicidal	Methanol	Leaves, roots and stems	Larval mortality of *Anopheles gambiae*	Extracts exhibited activities with stems exhibiting LC_50_ value of 139.1 µg/mL, roots (147.1 µg/mL) and leaves (240.4 µg/mL)	[2]
Cytotoxicity	Methanol	Roots	Plaque reduction assay	Extract showed weak activities on Vero cells with CC_50_ value of 225.3 µg/mL and selectivity index of 6.2 and 198.8 against W2 and D6, respectively	[102]
Cytotoxicity	Dichloromethane	Roots	Colorimetric assay via the oxidation-reduction indicator resazurin	Extract exhibited activities with IC_50_ value of 27.2 µg/mL and selectivity index of 1.2	[104]
Cytotoxicity	Methanol	Roots	Colorimetric assay via the oxidation-reduction indicator resazurin	Extract exhibited activities with IC_50_ value of 83.3 µg/mL and selectivity index of 1.9	[104]
Cytotoxicity	Aqueous	Stem bark	3-(4,5-dimethylthiazol-2-yl)-2,5-diphenyl tetrazolium bromide (MTT) calorimetric assay	Extract exhibited activities with CC50 values of >125 µg/mL	[97]
Cytotoxicity	80% methanol	Stem bark	MTT calorimetric assay	Extract exhibited activities with CC_50_ values of 72.3 µg/mL	[97]
Cytotoxicity	Aqueous	Leaves	Growth inhibition of Vero E6 cells	Extract exhibited mild activities with CC_50_ values of 76 µg/mL and a selectivity index of 1.4	[33]
Cytotoxicity	Methanol	Roots	Cytotoxicity determined against SH-SY5Y (human neuroblastoma) cells	Extract resulted in a concentration-dependent decrease in cell survival	[105]
Cytotoxicity	Methanol	Roots	MTT assay	Extract exhibited activities with IC_50_ values >100 µg/mL	[105]
Cytotoxicity	Methanol	Roots	Neutral red uptake assay	Extract exhibited activities with IC_50_ values >100 µg/mL	[105]
Cytotoxicity	Methanol	Roots	MTT assay	Extract exhibited activities with CC_50_ of 7.4 μg/mL and 74.0 μg/mL, on Vero cell lines and DU-145 prostate cancer cell lines, respectively	[100]
Toxicity	Methanol	Stems	Brine shrimp lethality test	Extract exhibited activities with LC_50_ value of 67.9 μg/mL	[58]
Toxicity	Methanol	Roots	in vivo acute toxicity activities	No deaths were observed at the highest concentration tested and LD_50_ values for both extracts was above 5000 mg/kg body weight	[102]

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
