# Peer review of "Review of Ethnomedicinal, Phytochemical and Pharmacological Properties of Lannea schweinfurthii (Engl.) Engl."

_molecules, 2019, doi:10.3390/molecules24040732_

Round 1

Reviewer 1 Report

It is an interesting and well documented review that suggest future studies and ideas for new drug discovery.

I suggest just minor language corrections.

Author Response

Responses are indicated in the attached pdf file

Reviewer 2 Report

The manuscript rewievs many studies on ethnomedicinal, phytochemical and pharmacological studies of the African plant Lannea schweinfurthii. This review may be useful for many researchers working in these fields.

The title is somewhat misleading as it may suggest original studies. It should be indicated that the paper is a review of original papers.

It is not a requirement but a not obligatory suggestion: could the authors include photo(s) or drawing(s) of the plant?

Abstract:

 „L. schweinfurthii is characterized by alkaloids, anthocyanins, flavonoids, glycosides, phenols, saponins, steroids, tannins and terpenoids”, please correct the sentence.

Fig. 1. Please extend the legend by information that countries in which Lannea schweinfurthii occurs are named.

Table 3. Please check the classification of compounds. E. g., malic acid is or qinic acid are not flavonoids.

Line 334: is ascorbic acid a drug?

Author Response

(The authors gave the same response as above.)
